# Impact of body and orofacial appearance on life satisfaction among Brazilian adults

**Lucas Arrais Campos**[1,2,3], **Juliana Alvares Duarte Bonini Campos**[4], **Wanderson Roberto da Silva**[4,5], **Timo Peltomäki**[1,2,6,7], **Ary dos Santos Pinto**[3], **João Marôco**[8]*

**1** Faculty of Medicine and Health Technology, Tampere University, Tampere, Finland, **2** Department of Ear and Oral Diseases, Tampere University Hospital, Tampere, Finland, **3** School of Dentistry, Campus Araraquara, São Paulo State University (UNESP), São Paulo, Brazil, **4** School of Pharmaceutical Sciences, São Paulo State University (UNESP), São Paulo, Brazil, **5** Postgraduate Program in Nutrition and Longevity, Alfenas Federal University, Minas Gerais, Brazil, **6** Faculty of Health Sciences, Institute of Dentistry, University of Eastern Finland, Kuopio, Finland, **7** Department of Oral and Maxillofacial Diseases, Kuopio University Hospital, Kuopio, Finland, **8** William James Center for Research (WJCR), ISPA-Instituto Universitário, Lisbon, Portugal

* jpmaroco@ispa.pt

**Data Availability Statement:** All relevant data are within the paper and its Supporting Information files.

## Abstract

### Aims

1. to elaborate a general model of physical appearance taking into account body image (BI) and orofacial appearance (OA) components; and 2. to evaluate the impact of BI and OA on life satisfaction among Brazilian adults.

### Methods

This is a cross-sectional observational study. The cognitive, behavioral, affective, and satisfaction/dissatisfaction aspects of BI, the satisfaction and psychosocial impact of OA, and life satisfaction were evaluated by self-reported psychometric scales. Principal Component Analysis and Parallel Analysis were performed. Structural equation models were elaborated to estimate the impact of BI and OA on life satisfaction. The fit of the models was verified and the significance of the path estimates (β) was evaluated using z-tests (α = 5%).

### Results

A total of 1,940 individuals participated in the study (age: mean = 24.8, standard deviation = 5.7 years; females = 70.1%). In the male sample, three physical appearance factors were retained (OA, cognitive and behavioral components of BI, and affective and satisfaction/dissatisfaction components of BI). In the female sample, two factors were retained (OA and all components of BI). All factors had significant impact on life satisfaction (β = |0.26|-|0.48|, p<0.001) in both samples. Individuals dissatisfied with BI and OA had lower levels of life satisfaction. For men, the affective and satisfaction components of BI had a greater impact on life satisfaction (β = 0.48, p<0.001) than the other factors (β =] -0.30;-0.25[, p<0.001). For women, both BI and OA had a similar impact (BI: β = -0.30, p<0.001; OA: β = -0.32, p<0.001).

**Funding:** This study was supported by the São Paulo Research Foundation (FAPESP) in the form a grant [2020/13153-3] and grants to WRdS [2017/20315-7], LAC [2018/06739-1], and JADBC [2019/19590-9], and by the William James Center for Research at ISPA, Instituto Universitário, in the form of funds to JM [UID/04810/2020]. This study was also supported in part by the Coordenação de Aperfeiçoamento de Pessoal de Nível Superior (CAPES), Brasil, in the form of funds to LAC [Finance Code 001]. The funders had no role in study design, data collection and analysis, decision to publish, or preparation of the manuscript.

**Competing interests:** The authors have declared that no competing interests exist.

## Conclusion

BI and OA formed distinct clusters in the physical appearance evaluation. Physical appearance was perceived differently by men and women, fostering discussion about the sociocultural construction of the body. BI and OA had a significant impact on life satisfaction and should be considered in assessment and treatment protocols.

## Introduction

Physical appearance plays an important role in social interactions. An observer often makes rapid inferences about another person by comparing the other's physical appearance with sociocultural standards and idealizations [1]. Because it is usually an important way for an individual to make themself visible to the world, physical appearance has a prominent place in people's lives. An individual can judge whether his/her physical characteristics are equivalent to internal or external expectations from a mentally formed picture of his/her own appearance [1]. If one's self-perception diverges from what one wants to look like, a state of dissatisfaction may arise. This can trigger behaviors such as, for example, changes in eating habits [2], consumption of substances to change the body [3], and the use of esthetic and surgical procedures [4]. Excessive concern with one's own physical appearance, as well as the compulsive adoption of these body-altering behaviors, are predictors of mental disorders such as eating disorders [2], body dysmorphic disorders [5], and anxiety [5].

The concept of body image is a mental representation of a person´s perceptions, thoughts and body feelings [6]. Body image is multidimensional and involves cognitive, affective, behavioral, and satisfaction/dissatisfaction aspects [6, 7]. Measurement of the body image is challenging, as it requires instruments capable of capturing body valuation from the individual's perception and/or attitudes [7]. Considering the multidimensionality of body image and the diversity of instruments available to measure it [8–15], future studies aiming to evaluate different body image components simultaneously have become increasingly relevant.

Attention paid to body shape (cognitive aspect) [8] and social physique anxiety (affective aspect) [9] are components of body image. The latter refers to anxiety generated by the perception of societal judgement of body appearance. These components can be assessed using psychometric scales, such as the Attention to Body Shape Scale (ABS) [8] and the Social Physique Anxiety Scale (SPAS) [9]. Body checking and avoidance are behavioral aspects of body image [10], which deal with reluctance to expose the body to oneself or others and the resources adopted to control and deprecate one's own body. They can be assessed by the Body Checking and Avoidance Questionnaire (BCAQ) [10]. Body Satisfaction Scale (BSS), with separate versions for women and men [12], measures individuals' satisfaction/dissatisfaction towards specific parts of their own body [11].

Orofacial appearance deserves attention since it has great importance in social insertion and in the construction of an individual's identity [16]. However, studies [8, 9, 14, 15, 17, 18] that evaluate body image usually do not include orofacial appearance among the investigated components. Inclusion of orofacial appearance could be relevant for a more comprehensive evaluation of body image. Orofacial appearance can also be measured by psychometric scales, the Orofacial Esthetics Scale (OES) [19] and the Psychosocial Impact of Dental Aesthetics Questionnaire (PIDAQ) [20]. OES assesses satisfaction with orofacial appearance [19] and PIDAQ assesses the impact that dental esthetics have on the individual's life, such as self-confidence, social impact, psychological impact, and concern about the appearance of teeth [20].

Once the individual's mental construction of their own body and orofacial appearance has been identified, it becomes possible to study its impact on well-being. Studies have concluded that positive body image components have a positive association with cognitive, emotional, social, and psychological well-being [17, 18, 21–23], whereas negative components, such as concerns with body shape, have a negative impact on well-being [21, 24, 25]. Regarding orofacial appearance, although it is one of the components of an individual's oral health experience [26] and is related to general health and well-being [27], there are still only a few studies which have investigated the direct impact of orofacial appearance on different aspects of well-being, such as life satisfaction.

Life satisfaction is a cognitive aspect of subjective well-being. It is a concept formed as a result of an individual's judgment of the comparison between her/his current circumstances and internalized standards [28, 29], and can be measured by the Satisfaction with Life Scale (SWLS) [28]. The literature [17, 18, 21–27] has pointed out an important impact of body image and orofacial appearance on life satisfaction, reinforcing the need for continuous and systematic studies that can provide evidence to contribute to clinal planning and decision making and to the advancement of science. The objectives of this study were 1. to elaborate a general model of physical appearance considering different components of body image (cognitive, affective, behavioral, and satisfaction/dissatisfaction aspects) and orofacial appearance (satisfaction and psychosocial impact), and 2. to evaluate the direct impact and the indirect effects of demographic characteristics and of body and orofacial appearance on life satisfaction among Brazilian adults.

## Methods

### Study design and sampling

This was a cross-sectional study with a convenience sample. Brazilian adult individuals of both sexes aged between 18 and 40 years were included in the study. Since body image and orofacial appearance may change throughout life [7, 30], the age range was limited to 40 years to minimize the effect of age on the results.

The minimum sample size was calculated following the proposal by Hair et al. [31], who recommended a minimum of 5 to 10 participants per observed variable to be included in the structural model. In the present study, 11 components related to physical appearance, 9 demographic variables, and 5 items from the SWLS were considered a priori to be included in the model to be tested, for a total of 25 observed variables. Thus, the minimum sample size was 125 to 250 participants. Since the concept of body image can be perceived differently between men and women [32, 33], the analyses were performed separately, which increased the requisite number of participants. In fact, a larger number of participants were recruited in order to increase the representativeness of the data for the study population.

### Procedures and ethical aspects

Data collection took place from August 2018 to December 2019. Initially, dental patients (in clinic waiting rooms), employees and students from the School of Dentistry of Araraquara (São Paulo State University–Unesp) were invited to participate in the study. Next, snowball sampling strategy was adopted to recruit more participants. For this purpose, after completing the data collection, each participant was asked to invite her/his family members and colleagues to participate in the study. The measuring instruments were self-filled using the paper-and-pencil method. First, the participants answered the demographic questions. The measuring instruments were then presented in random order to the different participants.

This study was approved by the Research Ethics Committee of São Paulo State University (Unesp), School of Dentistry, Araraquara (CAAE: 88600318.3.0000.5416). Only individuals who agreed with and signed the written Informed Consent participated in the study.

## Sample characterization

For sample characterization, the following demographic information (Table 1) was collected: age, gender (male, female), marital status (single, married/common law stable relationship, divorced, widower), economic status, whether the participant was undergoing any type of dental treatment at the time of participation (no, yes), had received any esthetic dental treatment (no, yes), had received orthodontic treatment (no, yes), liked her/his own smile (no, yes), whether the participant had ever undergone any surgical procedure exclusively to change bodily appearance, whether s/he had undergone any cosmetic procedures to improve bodily appearance (no, yes), body weight (kg), or height (m). Economic status was evaluated according to the Brazilian Economic Classification Criterion [34]. Individuals were classified as

**Table 1. Characterization of the total sample and the subsamples according to gender.**

| Characteristic | Male (n = 580) | Female (n = 1,360) | Total (n = 1,940) |
|---|---|---|---|
| **Age** (years, mean (95%CI)) | 25.0 (24.6–25.4) | 24.8 (24.5–25.1) | 24.8 (24.5–25.1) |
| **BMI**[*] (kg/m$^2$, mean (95%CI)) | 25.0 (24.6–25.4) | 23.9 (23.7–24.2) | 24.2 (24.0–24.5) |
| **Marital status** | n (95%CI) | | |
| Single | 499 (83.5–89.1) | 1,083 (77.8–82.0) | 1,582 (80.1–83.5) |
| Married/common law stable relationship | 76 (10.4–16.0) | 237 (15.4–19.4) | 313 (14.5–17.7) |
| Divorced | 3 (0.0–1.1) | 35 (1.8–3.4) | 38 (1.4–2.6) |
| Widower | - | 1 (0.0–0.3) | 1 (0.0–0.2) |
| **Economic Status** | | | |
| A | 155 (26.0–34.0) | 318 (24.2–29.2) | 473 (25.6–29.8) |
| B | 270 (47.9–56.5) | 644 (51.3–56.9) | 914 (51.1–55.9) |
| C | 90 (14.1–20.7) | 218 (16.1–20.5) | 308 (16.2–19.8) |
| D/E | 2 (0.0–0.9) | 11 (0.4–1.4) | 13 (0.4–1.2) |
| **Dental patient** | | | |
| No | 465 (77.0–83.4) | 999 (71.2–75.8) | 1,464 (73.6–77.4) |
| Yes | 115 (16.6–23.0) | 361 (24.2–28.8) | 476 (22.6–26.4) |
| **Have you received any esthetic dental treatment?** | | | |
| No | 206 (32.0–39.8) | 385 (26.1–30.9) | 591 (28.6–32.8) |
| Yes | 368 (60.2–68.0) | 966 (69.1–73.9) | 1,334 (67.2–71.4) |
| **Have you received any orthodontic treatment?** | | | |
| No | 311 (49.8–58.0) | 561 (38.9–44.3) | 872 (43.1–47.5) |
| Yes | 266 (42.0–50.2) | 789 (55.7–61.1) | 1,055 (52.5–56.9) |
| **Do you like your smile?** | | | |
| No | 135 (20.3–27.3) | 277 (18.5–22.9) | 412 (19.8–23.4) |
| Yes | 433 (72.7–79.7) | 1.063 (77.1–81.5) | 1,496 (76.6–80.2) |
| **Have you ever undergone a surgical procedure exclusively to change the appearance of your body?** | | | |
| No | 542 (91.9–95.9) | 1,182 (86.0–89.6) | 1,724 (88.2–91.0) |
| Yes | 35 (4.1–8.1) | 165 (10.4–14.0) | 200 (9.0–11.8) |
| **Have you undergone any cosmetic procedure to improve your body appearance?** | | | |
| No | 528 (89.4–94.0) | 1,144 (82.9–86.7) | 1,672 (85.4–88.4) |
| Yes | 48 (6.0–10.6) | 205 (13.3–17.1) | 253 (11.6–14.6) |

[*]BMI: body mass index

economic level D-E (mean monthly household income: R$813.56/U$149.25), C (R$1,805.91–3,042.47/U$331.30–558.15), B (R$5,449.60–10,427.74/U$999.74–1,913.00), or A (R$22,716.99/U$4,167.49). The values in US dollars (U$) were estimated from the Central Bank of Brazil quotation on October 15, 2021 (U$ 1.00 = R$ 5.45). Body weight and height reported by the participants were used to calculate the body mass index (BMI, $Kg/m^2$) and was considered a quantitative variable in the present study.

## Measuring scales

The ABS [8, 14] was used to assess the cognitive component of body image. It is a unifactorial scale composed of 7 items with a 5-point Likert-type response scale, ranging from 1 (definitely disagree) to 5 (definitely agree). When estimating the psychometric properties of this instrument with respect to the study sample, it was observed that one item (item 3: "I am not self-conscious about my body shape") did not have an adequate factor loading [14]. Therefore, this item was not considered for calculation of the mean score and analyses of the present study in order to avoid possible biases and ensure the validity and reliability of the estimates.

Social physical anxiety, an affective aspect of body image, was assessed using the SPAS [9, 35]. It contains 12 items with a 5-point Likert-type response scale (1: not at all characteristic of me, to 5: extremely characteristic of me) and measures two components of social physical anxiety: comfort with body presentation and expectation of negative physical evaluation [15]. The BCAQ [10, 36] was used to measure the behavioral component of body image. This scale is composed of 22 items with a 6-point Likert-type response scale (0: not at all/not interested; 1: checked less than once a week; 2: checked 1–6 times a week; 3: checked 1–2 times a day; 4: checked more than 3 times a day; 5: avoidance of checking because of possible distress). It measures 5 first-order factors (unnamed, since they represent avoidance and body-checking strategies and should not be interpreted as constructs) that reflect a second-order hierarchical factor (body checking and avoidance) [13], which allowed us to obtain a single score for this instrument in our study.

The component of dissatisfaction/satisfaction with body image was assessed by BSS [12]. This scale was originally developed in Portuguese [11] and contains 23 items with a 5-point Likert-type response scale (1: strongly disagree to 5: strongly agree) that assesses 4 factors: "dissatisfaction and fat", "external parts", "satisfaction and muscular condition", and "lower parts". However, a previous validation study indicated that this instrument worked differently between men and women [12]. Therefore, we used those factorial structures previously found for each sex [12]. For men, the components assessed were "satisfaction with body and muscles" and "satisfaction with external body parts", and for women, "dissatisfaction with body and fat" and "satisfaction with external body parts".

Components related to orofacial appearance were assessed using the OES [19, 37] and the PIDAQ [20, 38]. The OES is a one-factor scale that measures satisfaction with orofacial appearance and is composed of 7 items with an 11-point numerical response scale ranging from 0 (very dissatisfied) to 10 (very satisfied). For the PIDAQ, we used the version presented by Campos et al. [39] composed of 24 items that evaluate 4 factors of the psychosocial impact of dental esthetics: dental self-confidence, social impact, psychological impact, and esthetic concern. The response scale to the items is a 5-point Likert type (0: I do not agree to 4: I totally agree). The SWLS [28, 40] was used to measure the individual's overall life satisfaction. It consists of 5 items with a 7-point Likert-type response scale (1: strongly disagree to 7: strongly agree).

The psychometric properties of the measuring scales were attested for the sample data using Confirmatory Factor Analysis and Cronbach's alpha coefficient or ordinal alpha

coefficient (S1 Table). The fit of the scales to the samples was adequate, demonstrating the validity and reliability of the data obtained (S1 Table).

## Data analysis

The scores of the body and orofacial appearance components were calculated for each participant from the means of the answers given to the items of the scales. Participants who did not answer two or more items in at least one of the measurement scales were excluded from the analyses. Principal Component Analysis (PCA) with Promin rotation was conducted to explore a formative model of physical appearance using the component scores of body image and orofacial appearance. To verify the assumptions of this analysis, descriptive statistics were initially used to evaluate the approximation to the distribution of scores. Absolute values of skewness and kurtosis lower than 3 and 10, respectively, were considered indicative of no severe violation of normal distribution [41]. Then, the sampling adequacy for factoring was assessed by the measures of sampling adequacy (MSA). MSA values above 0.7 were considered acceptable [31]. A two-dimensional map was plotted considering the total sample (male + female), in order to verify the relative position of each individual according to gender in the common components for both sexes [42].

Parallel Analysis with random permutations of the observed data [43] was used to determine the number of factors to be retained. The suggestion of the unidimensionality of the dataset for each sample (male and female) was also evaluated considering the Unidimensional Congruence (UniCo), Explained Common Variance (ECV) and Mean of Item Residual Absolute Loadings (MIREAL) indices [44]. Values of UniCo > 0.95, ECV > 0.85 and MIREAL < 0.30 suggested that the scores can be treated as components of a single factor [44].

Once the models for each sex were defined, structural equation analysis was conducted to estimate the impact of physical appearance on life satisfaction. Separate structural models were elaborated for each factor retained in the parallel analysis, consisting of the different components of body and orofacial appearance. These components were considered as independent variables. The dependent variable was life satisfaction as assessed from the SWLS.

The variables 'being a dental patient' (0 = no, 1 = yes), 'having received any esthetic dental treatment' (0 = no, 1 = yes) and 'having received orthodontic treatment' (0 = no, 1 = yes) were inserted as intermediate variables (indirect effect) between orofacial appearance and life satisfaction. The variables 'having received surgical procedure exclusively to change body appearance' (0 = no, 1 = yes) and 'having undergone any cosmetic procedures to improve body appearance' (0 = no, 1 = yes) were inserted as intermediate variables between body image and life satisfaction. BMI (Kg/m$^2$) was considered as an independent variable impacting life satisfaction, with body image as an intermediate variable. The models elaborated from the results of the Parallel Analysis are shown in Fig 1. In order to verify the criteria for indirect effect, we followed the recommendations of Valeri and VanderWeele [45] and Baron and Kenny [46]. Bootstrap simulation analysis for Sobel's test was used for the evaluation of indirect effect path estimates [41]. In all structural models, economic status (1 = D/E, 2 = C, 3 = B, 4 = A) was also considered as an independent variable, since a previous study had pointed to a significant relationship between this variable and life satisfaction [47]. The goodness of fit of the models to the data was assessed using the reference values for the indices CFI≥0.90, TLI≥0.90, RMSEA≤0.10 and SRMR≤0.08 [41]. The significance of the hypothesized causal path estimates (β) was evaluated using the z-test (α = 5%).

PCA was performed using IBM SPSS Statistics 28 (IBM Corp., Armonk, NY, USA) and program Factor 11.05 for Windows (Ferrando and Lorenzo-Seva) [48], and structural equation modeling was performed in R program (R Core Team, 2020) using the *lavaan* [49] and *SEM-Tools* [50] packages.

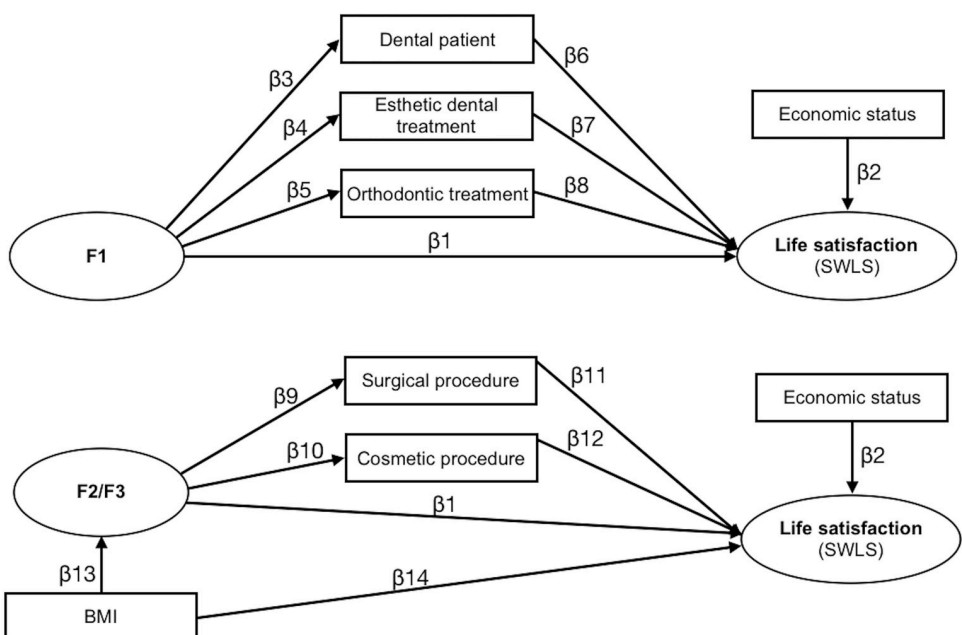

**Fig 1. Structural model elaborated to assess the impact of physical appearance factors, and the indirect effect of demographic characteristics, on life satisfaction.** Note. BMI: body mass index; F1, F2/F3: factors formed by the components retained in the sample according to gender as described earlier in the text (F1: orofacial appearance; F2/F3: body image).

## Results

A total of 2,154 individuals participated in the study. Of the participants, 214 were excluded because they did not answer two or more items from at least one of the measurement scales (male: n = 580; female: n = 1,360). Table 1 shows the characterization of the total sample and the used sample according to gender. Most of the participants were single, of medium-high economic status (A/B), were not dental patients, had already received some esthetics dental treatment, had already received orthodontic treatment, and had not undergone any surgical or esthetic procedures to change the appearance of their bodies. Some gender differences were observed. The female sample had a higher prevalence of dental patients who had already received esthetic dental treatment, orthodontic treatment, and surgical procedures, and who had undergone some esthetic procedure to change their body appearance.

The descriptive statistics of the scores for each component and MSA for principal component analysis are shown in Table 2. An approximation to normal distribution was observed. The MSA values obtained indicate that there is sampling adequacy for factoring, thus indicating that the data meet the assumptions for conducting the PCA.

On the basis of visual inspection of the biplot (S1 Fig), no gender differences were observed in the proximity of the individuals to the components. Values of the indices suggesting unidimensionality (male sample: UniCo = 0.87, EVC = 0.74, MIREAL = 0.34; female sample: UniCo = 0.81, EVC = 0.70, MIREAL = 0.37) do not provide support for the interpretation that the components can be treated essentially as a single dimension of body appearance. This finding is confirmed by the results of the PCA and Parallel Analysis (Table 3). The results suggest the retention of 3 factors for the male sample and 2 factors for the female sample, indicating that the components of physical appearance are interpreted differently between the sexes.

The loadings of the components retained in each factor are shown in Table 4. For the male sample, the component "expectation of negative physical evaluation" (SPAS) showed cross-

**Table 2. Descriptive statistics of scores of body image, orofacial appearance, and life satisfaction components assessed in the study and measures of sample adequacy (MSA) for principal component analysis (male sample: n = 580; female sample: n = 1,360).**

| Measuring Instrument Component* | Sample: male/female | | | | | | | |
|---|---|---|---|---|---|---|---|---|
| | Mean | Median | Standard deviation | Minimum | Maximum | Skewness | Kurtosis | MSA |
| **ABS** | | | | | | | | |
| Attention to Body Shape | 3.01/3.27 | 3.00/3.33 | 0.99/0.98 | 1/1 | 5/5 | 0.01/-0.22 | -0.75/-0.69 | 0.68/0.83 |
| **SPAS** | | | | | | | | |
| Comfort about body presentation | 2.93/2.55 | 3.00/2.50 | 0.89/0.89 | 1/1 | 5/5 | 0.14/0.29 | -0.24/-0.27 | 0.81/0.81 |
| Expectation of negative physical evaluation | 2.45/2.89 | 2.29/2.86 | 1.06/1.07 | 1/1 | 5/5 | 0.59/0.19 | -0.57/-0.95 | 0.82/0.86 |
| **BCAQ** | | | | | | | | |
| Body checking and avoidance | 0.86/1.16 | 0.73/1.00 | 0.67/0.78 | 0/0 | 5/5 | 1.49/1.19 | 3.32/1.67 | 0.81/0.88 |
| **BSS** | | | | | | | | |
| Satisfaction with body and muscles # | 3.08/- | 3.00/- | 1.02/- | 1/- | 5/- | 0.09/- | -0.76/- | 0.79/- |
| Dissatisfaction with body and fat † | -/3.20 | -/3.20 | -/1.26 | -/1 | -/5 | -/-0.10 | -/-1.26 | -/0.79 |
| Satisfaction with external body parts | 3.68/3.51 | 3.75/3.50 | 0.98/1.00 | 1/1 | 5/5 | -0.55/-0.41 | -0.19/-0.45 | 0.88/0.85 |
| **PIDAQ** | | | | | | | | |
| Dental self-confidence | 2.04/2.06 | 2.00/2.14 | 1.03/1.09 | 0/0 | 4/4 | 0.01/-0.06 | -0.96/-0.98 | 0.81/0.81 |
| Social impact | 0.63/0.68 | 0.25/0.38 | 0.78/0.88 | 0/0 | 4/4 | 1.64/1.67 | 2.70/2.35 | 0.79/0.82 |
| Psychological impact | 0.94/1.00 | 0.67/0.73 | 0.88/0.95 | 0/0 | 4/4 | 1.24/1.10 | 1.22/0.59 | 0.83/0.83 |
| Esthetic concern | 0.86/0.88 | 0.67/0.67 | 0.96/1.03 | 0/0 | 4/4 | 1.13/1.21 | 0.77/0.75 | 0.90/0.91 |
| **OES** | | | | | | | | |
| Satisfaction with Orofacial Appearance | 6.94/6.72 | 7.07/7.07 | 1.72/2.03 | 0/0 | 10/10 | -0.73/-0.75 | 1.27/0.31 | 0.84/0.85 |
| **SWLS** | | | | | | | | |
| Life satisfaction | 4.79/4.79 | 5.00/5.00 | 1.30/1.30 | 1/1 | 7/7 | -0.58/-0.60 | -0.11/-0.11 | -/- |

*ABS: Attention to Body Shape Scale; SPAS: Social Physique Anxiety Scale; BCAQ: Body Checking and Avoidance Questionnaire; BSS: Body Satisfaction Scale; PIDAQ: Psychosocial Impact of Dental Aesthetics Questionnaire; OES: Orofacial Esthetics Scale; SWLS: Satisfaction with Life Scale. # Dimension measured only in the male subsample. † Dimension measured only in the female subsample

**Table 3. Principal component analysis (PCA) and parallel analysis results for the male (n = 580) and female (n = 1,360) samples.**

| Dimensions | Male Sample | | | | Female Sample | | | |
|---|---|---|---|---|---|---|---|---|
| | PCA | | Parallel Analysis | | PCA | | Parallel Analysis | |
| | Real-data eigenvalues | Proportion of variance | Mean of random eigenvalues | 95 percentile of random eigenvalues | Real-data eigenvalues | Proportion of variance | Mean of random eigenvalues | 95 percentile of random eigenvalues |
| 1 | 4.29* | 38.98 | 1.23 | 1.28 | 4.28* | 38.92 | 1.14 | 1.18 |
| 2 | 1.76* | 15.99 | 1.16 | 1.20 | 2.12* | 19.24 | 1.10 | 1.13 |
| 3 | 1.20* | 10.94 | 1.11 | 1.15 | 1.02 | 9.25 | 1.07 | 1.10 |
| 4 | 0.80 | 7.29 | 1.07 | 1.10 | 0.79 | 7.16 | 1.05 | 1.07 |
| 5 | 0.74 | 6.74 | 1.03 | 1.06 | 0.65 | 5.93 | 1.02 | 1.04 |
| 6 | 0.57 | 5.21 | 1.00 | 1.02 | 0.57 | 5.20 | 1.00 | 1.02 |
| 7 | 0.46 | 4.14 | 0.96 | 0.99 | 0.43 | 3.90 | 0.97 | 0.99 |
| 8 | 0.38 | 3.46 | 0.92 | 0.95 | 0.37 | 3.37 | 0.95 | 0.97 |
| 9 | 0.33 | 2.96 | 0.88 | 0.92 | 0.32 | 2.93 | 0.92 | 0.94 |
| 10 | 0.29 | 2.66 | 0.84 | 0.88 | 0.28 | 2.54 | 0.90 | 0.92 |
| 11 | 0.18 | 1.63 | 0.79 | 0.83 | 0.17 | 1.57 | 0.86 | 0.89 |

*Suggesting components to be retained by Parallel Analysis (real-data eigenvalues >mean of random eigenvalues)

**Table 4. Loadings of the components included in the principal component analysis for the male (n = 580) and female (n = 1,360) samples.**

| Components* | Male Sample | | | | | | Female Sample | |
|---|---|---|---|---|---|---|---|---|
| | 3 factors retained | | | 3 factors retained–refined# | | | 2 factors retained | |
| | F1 | F2 | F3 | F1 | F2 | F3 | F1 | F2 |
| ABS | -0.02 | 0.05 | **0.80** | -0.01 | 0.08 | **0.83** | -0.23 | **0.70** |
| SPAS_F1 | 0.07 | **0.79** | -0.28 | 0.04 | **0.77** | -0.24 | -0.05 | **-0.73** |
| SPAS_F2 | 0.04 | **-0.51** | **0.56** | - | - | - | 0.07 | **0.77** |
| BCAQ | 0.01 | -0.12 | **0.77** | 0.01 | -0.11 | **0.80** | -0.01 | **0.70** |
| BSS_F1 | 0.05 | **0.85** | -0.12 | 0.04 | **0.84** | -0.09 | -0.09 | **0.85** |
| BSS_F2 | -0.13 | **0.57** | 0.01 | -0.11 | **0.61** | -0.02 | **-0.38** | -0.08 |
| PIDAQ_F1 | **-0.73** | 0.11 | 0.20 | **-0.72** | 0.13 | 0.19 | **-0.82** | 0.08 |
| PIDAQ_F2 | **0.84** | 0.09 | 0.16 | **0.86** | 0.11 | 0.13 | **0.81** | 0.05 |
| PIDAQ_F3 | **0.91** | 0.05 | 0.06 | **0.91** | 0.06 | 0.05 | **0.87** | 0.06 |
| PIDAQ_F4 | **0.88** | 0.18 | 0.08 | **0.88** | 0.18 | 0.08 | **0.82** | -0.05 |
| OES | **-0.70** | 0.38 | 0.14 | **-0.68** | 0.30 | 0.12 | **-0.82** | -0.02 |

*ABS: attention to body shape measured by Attention to Body Shape scale; SPAS_F1: comfort about body presentation measured by the Social Physique Anxiety Scale (SPAS); SPAS_F2: expectation of negative physical evaluation measured by SPAS; BCAQ: body checking and avoidance assessed by Body Checking and Avoidance Questionnaire; BSS_F1: satisfaction with body and muscles (male sample) or dissatisfaction with body and fat (female sample) measured by the Body Satisfaction Scale (BSS); BSS_F2: satisfaction with external body parts measured by BSS; PIDAQ_F1: dental self-confidence measured by Psychosocial Impact of Dental Aesthetics Questionnaire (PIDAQ); PIDAQ_F2: social impact measured by PIDAQ; PIDAQ_F3: psychological impact measured by PIDAQ; PIDAQ_F4: esthetic concern measured by PIDAQ; OES: satisfaction with orofacial appearance measured by Orofacial Esthetics Scale. #Exclusion of the component "negative physical evaluation expectation"

loading between two of the retained factors, so the model was refined by its exclusion. Thus, for the male sample, it was observed that the components measured by the instruments related to orofacial appearance (PIDAQ and OES) were allocated to the same factor (F1). The body image-related components were allocated to two separate factors, one containing those measured by the SPAS and the BSS (F2) and the other those measured by the ABS and BCAQ (F3). The correlation (r) between the factors ranged from low to moderate ($r_{F1xF2}$ = -0.37, p<0.001; $r_{F1xF3}$ = 0.20, p<0.001; $r_{F2xF3}$ = -0.14, p<0.001).

For the female sample, two factors were retained. The orofacial appearance-related components were allocated to one factor (F1) and those related to body image to another (F2), except for the component "satisfaction with external body parts", measured by BSS, which was allocated to the orofacial appearance factor (F1). However, it should be noted that the loading of this last component was lower than those found in the others present in this factor. The correlation between these two factors was $r_{F1xF2}$ = 0.35, p<0.001.

Fig 1 shows a representation of the structural models elaborated to assess the impact of physical appearance on life satisfaction, and the analyses are shown in Table 5. In all the models tested, it was observed that there was no indirect effect of demographic variables, so the models were refined by the exclusion of these variables.

For both the male and female samples, physical appearance factors had a significant impact on life satisfaction (Table 6). The higher the satisfaction with orofacial appearance and the lower the psychosocial impact of dental esthetics, the higher was the life satisfaction. For body image, the higher the attention to body shape, dissatisfaction with body and fat, expectation of negative physical evaluation, and more avoidance and checking behaviors, the lower was the life satisfaction, whereas individuals with higher levels of satisfaction with body and muscles and comfort about body presentation had higher life satisfaction.

**Table 5. Path estimates of the structural models elaborated to assess the impact of physical appearance factors, and the indirect effect of demographic characteristics, on life satisfaction.**

| Path estimate | Male sample | | | | Female sample | | | |
|---|---|---|---|---|---|---|---|---|
| | B | β | SE | p | B | β | SE | p |
| **IV: F1†** | | | | | | | | |
| IV → SWLS (β1) | -0.577 | -0.322 | 0.101 | <0.001** | -0.555 | -0.336 | 0.066 | <0.001** |
| ES → SWLS (β2) | 0.167 | 0.102 | 0.080 | 0.038** | 0.321 | 0.182 | 0.055 | <0.001** |
| IV → Patient (β3) | 0.094 | 0.149 | 0.033 | 0.005** | 0.096 | 0.161 | 0.020 | <0.001** |
| IV → Aesth.T. (β4) | -0.040 | -0.053 | 0.038 | 0.287 | -0.053 | -0.087 | 0.020 | 0.009** |
| IV → Ortho.T. (β5) | -0.032 | -0.040 | 0.040 | 0.424 | -0.015 | -0.022 | 0.022 | 0.494 |
| Patient → SWLS (β6) | 0.245 | 0.087 | 0.132 | 0.064 | -0.018 | -0.007 | 0.085 | 0.830 |
| Esth.T. → SWLS (7) | 0.125 | 0.053 | 0.145 | 0.387 | 0.153 | 0.056 | 0.108 | 0.156 |
| Ortho.T. → SWLS (β8) | 0.002 | 0.001 | 0.136 | 0.987 | -0.006 | -0.002 | 0.094 | 0.951 |
| **Indirect effect#** | | | | | | | | |
| IV→Patient→SWLS (β3*β6) | 0.023 | 0.013 | 0.015 | 0.126 | -0.002 | -0.001 | 0.008 | 0.834 |
| IV→Esth.T.→SWLS (β4*β7) | -0.005 | -0.003 | 0.009 | 0.582 | -0.008 | -0.005 | 0.007 | 0.237 |
| IV→Ortho.T.→SWLS (β5*β8) | 0.000 | 0.000 | 0.007 | 0.992 | 0.000 | 0.000 | 0.002 | 0.972 |
| **IV: F2†** | | | | | | | | |
| IV → SWLS (β1) | 0.838 | 0.489 | 0.123 | <0.001** | -0.912 | -0.336 | 0.132 | <0.001** |
| ES → SWLS (β2) | 0.218 | 0.132 | 0.082 | 0.008** | 0.483 | 0.268 | 0.056 | <0.001** |
| IV → Surgic.Pro. (β9) | -0.009 | -0.024 | 0.014 | 0.525 | 0.017 | 0.024 | 0.022 | 0.437 |
| IV → Cosm.Pro. (β10) | 0.018 | 0.044 | 0.020 | 0.379 | 0.029 | 0.037 | 0.025 | 0.249 |
| Surgic.Pro. → SWLS (β11) | -0.119 | -0.026 | 0.235 | 0.611 | -0.030 | -0.008 | 0.111 | 0.788 |
| Cosm.Pro.→ SWLS (β12) | -0.176 | -0.042 | 0.228 | 0.439 | 0.194 | 0.056 | 0.096 | 0.042** |
| IMC → IV (β13) | -0.030 | -0.217 | 0.008 | <0.001** | 0.043 | 0.424 | 0.004 | <0.001** |
| IMC → SWLS (β14) | 0.006 | 0.024 | 0.011 | 0.619 | 0.014 | 0.051 | 0.010 | 0.155 |
| **Indirect effect #** | | | | | | | | |
| IV→Surgic.Pro.→ SWLS (β9*β11) | 0.001 | 0.001 | 0.004 | 0.806 | -0.001 | 0.000 | 0.003 | 0.868 |
| IV→Cosm.Pro.→ SWLS (β10*β12) | -0.003 | -0.002 | 0.008 | 0.967 | 0.006 | 0.002 | 0.006 | 0.356 |
| IMC→IV→SWLS (β13*β14) | -0.025 | -0.106 | 0.007 | <0.001** | -0.039 | -0.143 | 0.006 | <0.001** |
| **IV: F3†** | | | | | | | | |
| IV → SWLS (β1) | -0.585 | -0.270 | 0.148 | <0.001** | - | - | - | - |
| ES → SWLS (β2) | 0.262 | 0.160 | 0.088 | 0.003** | - | - | - | - |
| IV → Surgic.Pro. (β9) | -0.006 | -0.012 | 0.023 | 0.811 | - | - | - | - |
| IV → Cosm.Pro. (β10) | 0.023 | 0.045 | 0.025 | 0.351 | - | - | - | - |
| Surgic.Pro. → SWLS (β11) | -0.196 | -0.042 | 0.232 | 0.399 | - | - | - | - |
| Cosm.Pro.→ SWLS (β12) | -0.049 | -0.012 | 0.211 | 0.815 | - | - | - | - |
| IMC → IV (β13) | 0.012 | 0.111 | 0.008 | 0.119 | - | - | - | - |
| IMC → SWLS (β14) | -0.012 | -0.050 | 0.013 | 0.354 | - | - | - | - |
| **Indirect effect #** | | | | | | | | |
| IV→Surgic.Pro.→SWLS (β9*β11) | 0.001 | 0.001 | 0.007 | 0.879 | - | - | - | - |
| IV→Cosm.Pro.→SWLS (β10*β12) | -0.001 | -0.001 | 0.007 | 0.872 | - | - | - | - |
| IMC→IV→SWLS (β13*β14) | -0.007 | -0.030 | 0.004 | 0.109 | - | - | - | - |

B: non-standardized path estimate; β: standardized path estimate; SE: standard error; IV: independent variable; ES: economic status; SWLS: life satisfaction; Patient: dental patient (0 = no, 1 = yes); Esth.T.: esthetic dental treatment (0 = no, 1 = yes); Ortho.T.: orthodontic treatment (0 = no, 1 = yes); Surgic.Pro.: surgical procedure to change body appearance (0 = no, 1 = yes); Cosm.Pro.: cosmetic procedures to improve body appearance(0 = no, 1 = yes); BMI: body mass index.

†F1: orofacial appearance; F2/F3: body image. F1, F2, and F3: factors formed by the components retained for sample of each gender as described earlier in the text.

#Indirect effect assessed by Sobel's test with bootstrap simulation.

**p<0.05. β1 to β14: path estimates corresponding to Fig 1.

**Table 6. Fit to data and path estimates of the refined structural models (male and female samples) elaborated to assess the impact of physical appearance factors on life satisfaction.**

| Factor† | CFI | TLI | RMSEA | SRMR | Factor → SWLS (β1) | | | | ES → SWLS (β2) | | | | BMI → F2 (β13) | | | | EV |
|---|---|---|---|---|---|---|---|---|---|---|---|---|---|---|---|---|---|
| | | | | | B | β | SE | p | B | β | SE | p | B | β | SE | p | |
| **Male sample** | | | | | | | | | | | | | | | | | |
| F1# | 0.94 | 0.92 | 0.08 | 0.08 | -0.533 | -0.299 | 0.088 | <0.001 | 0.168 | 0.102 | 0.075 | 0.025 | - | - | - | - | 0.099 |
| F2 | 0.94 | 0.92 | 0.07 | 0.05 | 0.838 | 0.484 | 0.105 | <0.001 | 0.223 | 0.135 | 0.073 | 0.002 | -0.029 | -0.214 | 0.007 | <0.001 | 0.251 |
| F3 | 0.96 | 0.94 | 0.07 | 0.04 | -0.572 | -0.258 | 0.133 | <0.001 | 0.257 | 0.155 | 0.076 | 0.001 | - | - | - | - | 0.090 |
| **Female sample** | | | | | | | | | | | | | | | | | |
| F1# | 0.95 | 0.93 | 0.08 | 0.08 | -0.528 | -0.323 | 0.052 | <0.001 | 0.349 | 0.197 | 0.052 | <0.001 | - | - | - | - | 0.143 |
| F2 | 0.93 | 0.91 | 0.08 | 0.06 | -0.833 | -0.304 | 0.102 | <0.001 | 0.491 | 0.271 | 0.054 | <0.001 | 0.043 | 0.422 | 0.004 | <0.001 | 0.170 |

CFI: *Comparative Fit Index*; TLI: *Tucker-Lewis Index*; SRMR: *Standardized Root Mean Square Residual*; B: non-standardized path estimate; β: standardized path estimate; SE: standard error; ES: economic status; SWLS: life satisfaction assessed by the Satisfaction with Life Scale; EV: explained variance for SWL.

†F1: orofacial appearance; F2/F3: body image. F1, F2, and F3: factors formed by the components retained for samples of each gender as described earlier in the text.

#Correlation inserted between the errors of the components "satisfaction with orofacial appearance" and "dental self-confidence": male sample: r = 0.50; female: r = 0.49; β1, β2 and β13: path estimates corresponding to Fig 1.

It was observed in the male sample that the model with the cognitive and behavioral components of body image (F3) showed the lowest explained variance for life satisfaction (9.0%), whereas the model with the affective and satisfaction components (F2) showed significantly higher explained variance (25.1%). The factor containing the components of orofacial appearance showed an explained variance of 9.9% in this sample. In the female sample, the models containing the components of orofacial appearance (F1) and body appearance (F2) showed similar explained variances for life satisfaction (14.3% and 17.0%, respectively). Furthermore, for both samples, BMI did not have a direct impact on life satisfaction, but had a significant impact on the factor containing body image components (F2).

For men, the higher the BMI, the lower was the comfort with body presentation and satisfaction with body and muscles. For women, the higher the BMI, the greater was the attention to body shape, dissatisfaction with body and fat, expectation of negative physical evaluation, avoidance and checking behavior, and the lower was comfort with body presentation.

## Discussion

This study presents physical appearance models for both men and women, considering components related to body and orofacial image. Although these two images have different definitions [51], a model that includes both may allow for a more holistic understanding about the image an individual has of her/his physical appearance. It can broaden discussions and provide theoretical subsidies to researchers and professionals in the field, allowing decision making and clinical management to better target the needs of each individual. The present study also investigated the impact of physical appearance on life satisfaction and we found that both body image and orofacial appearance had a significant impact on this satisfaction.

One result observed in both the male and female samples was that in order to obtain the physical appearance model, the orofacial appearance components were grouped into a different factor than those related to body image. One reason for this could be the content of the instruments used in the present study, which involve specific aspects and components of either the face or the body. It is also noteworthy that we used two instruments assessing satisfaction, one related to the body (BSS) and the other to the face (OES), and that even so, body and face were allocated to different groupings. This corroborates the findings of Frederick et al. [52],

who observed that the facial components were grouped together in a different factor from the other body components when evaluating satisfaction with parts of the body in a sample of the North American population. We can suggest that the face is interpreted differently from other body components by individuals and may be related to psychosocial aspects. The face has a privileged place in the process of social interaction and in the constitution of an individual's identity [53], and is considered to be the richest and most powerful tool for non-verbal communication [54]. Therefore, these attributes of the face can differentiate it from other body components.

Although body image and orofacial appearance form distinct clusters, we emphasize that the clinical evaluation of both by health professionals can be relevant, especially if the patient brings an exclusively esthetic demand. This assessment, together with good communication between professional and patient, can provide clues as to whether the demand for the procedure is due to a dissatisfaction specific to a physical aspect, or is an overvalued and/or generalized dissatisfaction of physical appearance [55]. In the first scenario, the esthetic procedure may be sufficient to meet the patient's expectations. The second situation, on the other hand, should serve as a warning to the professional to balance the risks and benefits of the procedure, since the patient may have body dysmorphia disorder or more generalized symptoms, such as anxiety. In such cases, patients may even experience a momentary satisfaction with the esthetic procedure; however, this may not be enough to bring them long-term benefits and satisfaction [5, 55]. Thus, the professional should have the ability to suspect and identify possible patients with these conditions and to refer them to specialized professionals, such as psychologists and psychiatrists, for diagnosis and the elaboration of an adequate follow-up and/or treatment plan.

Differences in body image between men and women were also found. For women it was observed that the cognitive, behavioral, affective, and satisfaction/dissatisfaction components were grouped into a single factor. This indicates that their cognitive-behavioral investment in physical appearance is related to their emotions and self-evaluation of physical appearance. In the case of the men, it was observed that the body image components were split into two factors having low correlation with each other (r = -0.14). In one grouping were the cognitive and behavioral components, and in the other the affective and satisfaction/dissatisfaction components. Given these results, for men the emotional process attributed to body appearance may be disconnected from the cognition and behavior spent on this appearance. This difference between men and women can be explained by a biopsychosocial developmental model of emotional expression. Chaplin [56] proposes the existence of social rules, according to which girls are expected to display positive emotions and internalize negative emotions, whereas boys are encouraged to limit and show less of these emotions. Once these social rules and behaviors related to emotions have been learned in childhood, they become a constituent part of an individual. Thus, adults will reproduce these social rules when facing different situations and aspects of life and in their perceptions, such as that of their own physical appearance. Therefore, it is suggested that for men, affect takes second place, while cognition and behavior have more relevance in the construction of their body image.

In addition, the difference between men and women can also be explained by the objectification theory [57]. This theory proposes the existence of a sociocultural construction in which the female body is seen as an object to be observed and judged by others [7, 57]. Because of this influence, girls learn self-objectification from childhood [7, 57]. They start to pay attention to their bodies and make a self-evaluation based on comparison with socially established beauty standards. However, these beauty standards have become increasingly illusory and unattainable for a significant portion of the population [58], which makes women feel dissatisfied [58]. They then adopt behaviors that aim to modify their body and appearance so that, in

their own judgment, they become more pleasing in the eyes of others, in order to avoid negative external judgments [7, 57, 58]. The process of self-objectification, which involves cognition, emotion, satisfaction/dissatisfaction, and behavior, serves as a mold for the construction of women's personal identity. Therefore, the construction of body image may involve the interaction of all these aspects in women of any age, especially in adolescence and the age group (18–40 years) of the present study sample. Following this sociocultural construction, for men this process is different, since from childhood they are discouraged from developing cognitive and behavioral investment in their body [7]. Although both theories (the biopsychosocial developmental model of emotional expression and the objectification theory) are sufficient to justify the present findings, we emphasize that in the present study only the attitudinal components of the cognitive-behavioral theoretical model of body image proposed by Cash [7] were evaluated. Thus, future studies that simultaneously assess the components of the expression of emotions and self-objectification and the attitudinal and perceptual components of body image may be relevant to obtaining empirical evidence about the relationship between them and possible justifications for body image differences between men and women.

Regarding the impact of body and orofacial appearance components on life satisfaction, the findings of the present study corroborate previous studies [21–23, 25]. Individuals with higher levels of negative aspects related to physical appearance (e.g., dissatisfaction) presented lower levels of life satisfaction, whereas individuals with higher levels of positive aspects related to appearance (e.g., satisfaction) experienced higher levels of satisfaction. Furthermore, physical appearance factors, either body- or face-related, explained from 9 to 25% of the variance in life satisfaction. Importantly, this satisfaction is a component of the individual's overall subjective well-being [28, 29]. Thus, identifying a single factor contributing 9% or more to this factor becomes relevant in the consideration of health actions aiming to promote an individual's well-being. It is therefore suggested that physical appearance be carefully evaluated so that these actions can be individualized in order to obtain better results according to the needs and demands of each person.

It was also observed in the structural models that BMI had no direct impact on life satisfaction. By contrast, Swami et al. [17] observed that BMI showed a weak but significant negative correlation with emotional, social, and psychological well-being in British men and women. This discrepancy between the results may have been due firstly to sampling differences between the studies. Secondly, Swami et al. [17] used a scale (Short Form of the Mental Health Continuum) that evaluates well-being from a eudaimonic perspective (focus on experiences of meaning and purpose), whereas the present study used an instrument that evaluates one of the components of well-being from a hedonic perspective (focus on experiences of pleasure and joy). What was observed in the present study, in consensus with others [17, 18, 21], was the impact of BMI on aspects of body image. Individuals with higher BMI values had higher levels of the negative components of body image and lower levels of the positive components. These results indicate that body image attitudes may be related to beauty standards and ideas, which in Western cultures refer to thin women and thin or athletically built men [18].

Limitations of the study include its cross-sectional design, which does not allow for the inference of cause and effect between the variables included in the structural models. The convenience sampling design, in which we adopted a snowball strategy restricted to the southeast region of Brazil, can also be pointed out as a limitation. This strategy limits the generalizability of the results to the Brazilian population. Brazil has a wide continental territory with marked sociocultural differences, which could affect body and face image, as well as economic differences, thus also affecting life satisfaction [47] in its different regions. However, although future studies including representative samples of Brazilian regions are needed, we believe that no substantial differences will be found, since the theoretical background [7, 56, 57] used in our

study refers to Western cultures, even with the variation existing among them. Despite these limitations, the strengths of the study include the analysis methods, the validity of the data obtained by the scales, which was attested to ensure the quality of the evidence presented, and the elaboration of structural models using robust techniques.

Thus, we expect that the results of this study could help health care professionals to develop and improve a holistic view of human subjects, which would assist the development of patient-centered treatment plans. It is suggested that a detailed anamnesis be performed, including psychometric scales (such as those used in the present study) and a question about how many aesthetic treatments the patient has had previously [5]. In this regard, it is also important that curricula for training courses and continuing education programs are steadily enhanced, updating current social issues and new scientific evidence and methodology located at the frontier of different fields of knowledge. Hence, it is possible to train health care professionals who can properly interpret clinical findings (including those obtaining by psychometric scales), who are aware of and can deal with the complexity of the biopsychosocial aspects involved in a treatment demand, and who assume the social role of their profession.

We also expect these results to contribute to research in the field by providing new evidence concerning the different aspects and components related to an individual's own view of their physical appearance, paving the way for future studies to expand this discussion. We also suggest that new measuring scales be developed considering the face and body components simultaneously in order to deepen knowledge and establish new directions toward individual and collective well-being.

## Conclusion

The construction of the image of one's own physical appearance was different between men and women. Men, unlike women, had body image-related cognitive-behavioral aspects disconnected from emotional ones, thus fostering discussion and reflection about the sociocultural constructs involved in valuing physical appearance. Both body and orofacial appearance had a significant impact on life satisfaction, and therefore it is recommended that these be included in assessment and treatment protocols, especially those involving an esthetic requirement.

## Supporting information

**S1 Table. Psychometric properties of the instruments fitted to the study samples.** (DOCX)

**S1 Fig. Two-dimensional map (biplot) containing the relative position of each participant, according to gender, with respect to the components of body image and orofacial appearance.** M: male; F: female; ABS: attention to body shape measured by Attention to Body Shape scale; SPAS_F1: comfort about body presentation measured by the Social Physique Anxiety Scale (SPAS); SPAS_F2: expectation of negative physical evaluation measured by SPAS; BCAQ: body checking and avoidance assessed by the Body Checking and Avoidance Questionnaire; ESSC_F2: satisfaction with external body parts measured by the Body Satisfaction Scale; PIDAQ_F1: dental self-confidence measured by the Psychosocial Impact of Dental Aesthetics Questionnaire (PIDAQ); PIDAQ_F2: social impact measured by PIDAQ; PIDAQ_F3: psychological impact measured by PIDAQ; PIDAQ_F4: esthetic concern measured by PIDAQ; OES: satisfaction with orofacial appearance measured by the Orofacial Esthetics Scale. (TIF)

**S1 File Data. Data underlying the finds described in this manuscript.** (XLSX)

## Acknowledgments

The authors would like to thank Arthur Fiorin Ragazzi for his collaboration in collecting data.

## Author Contributions

**Conceptualization:** Lucas Arrais Campos, Juliana Alvares Duarte Bonini Campos, Wanderson Roberto da Silva, Ary dos Santos Pinto, João Marôco.

**Data curation:** Lucas Arrais Campos, Juliana Alvares Duarte Bonini Campos, Wanderson Roberto da Silva.

**Formal analysis:** Lucas Arrais Campos, Juliana Alvares Duarte Bonini Campos, João Marôco.

**Funding acquisition:** Lucas Arrais Campos, Juliana Alvares Duarte Bonini Campos, Wanderson Roberto da Silva, Timo Peltomäki, João Marôco.

**Investigation:** Lucas Arrais Campos, Juliana Alvares Duarte Bonini Campos, Wanderson Roberto da Silva.

**Methodology:** Lucas Arrais Campos, Juliana Alvares Duarte Bonini Campos, Wanderson Roberto da Silva, Timo Peltomäki, João Marôco.

**Project administration:** Juliana Alvares Duarte Bonini Campos, João Marôco.

**Resources:** Lucas Arrais Campos, Juliana Alvares Duarte Bonini Campos, Wanderson Roberto da Silva, Ary dos Santos Pinto.

**Software:** Juliana Alvares Duarte Bonini Campos, João Marôco.

**Supervision:** Juliana Alvares Duarte Bonini Campos, Timo Peltomäki, Ary dos Santos Pinto, João Marôco.

**Validation:** Lucas Arrais Campos, Juliana Alvares Duarte Bonini Campos, Wanderson Roberto da Silva.

**Visualization:** Lucas Arrais Campos, Timo Peltomäki, João Marôco.

**Writing – original draft:** Lucas Arrais Campos, Juliana Alvares Duarte Bonini Campos.

**Writing – review & editing:** Wanderson Roberto da Silva, Timo Peltomäki, Ary dos Santos Pinto, João Marôco.

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
