## [Decision Letter · Decision Letter 0]

28 Jul 2022

PONE-D-22-14345Impact of body and orofacial appearance on life satisfaction among Brazilian adultsPLOS ONE

Dear Dr. Marôco,

Thank you for submitting your manuscript to PLOS ONE. After careful consideration, we feel that it has merit but does not fully meet PLOS ONE’s publication criteria as it currently stands. Therefore, we invite you to submit a revised version of the manuscript that addresses the points raised during the review process.

We look forward to receiving your revised manuscript.

Kind regards,

Manoelito Ferreira Silva Junior, Ph.D.

Academic Editor

PLOS ONE

Journal Requirements:

"This study received financial support from grants #2018/06739-1, #2019/19590-9, and #2020/13153-3, São Paulo Research Foundation (FAPESP). This study was financed in part by the Coordenação de Aperfeiçoamento de Pessoal de Nível Superior - Brasil (CAPES) – Finance Code 001."

Additional Editor Comments:

Dear authors,

I encourage them to make the corrections requested by the reviewers, for the smooth running of the editorial process of the manuscript.

Best Regards.

Reviewers' comments:

Reviewer's Responses to Questions

**Comments to the Author**

1. Is the manuscript technically sound, and do the data support the conclusions?

Reviewer #1: Yes

Reviewer #2: Yes

2. Has the statistical analysis been performed appropriately and rigorously? 

Reviewer #1: Yes

Reviewer #2: Yes

3. Have the authors made all data underlying the findings in their manuscript fully available?

Reviewer #1: Yes

Reviewer #2: Yes

4. Is the manuscript presented in an intelligible fashion and written in standard English?

Reviewer #1: No

Reviewer #2: Yes

5. Review Comments to the Author

Reviewer #1: Manuscript: Impact of body and orofacial appearance on life satisfaction among Brazilian adults

Abstract

Authors should it clear which form of evaluation, if it occurred through questionnaires.

In the results, the authors should add the real value of p in the sentence: “For men, the affective and satisfaction components of BI had a greater impact on life satisfaction than the other factors. For women, both BI and OA had a similar impact.”

Introduction

The introduction is too long.

“…studies aiming to evaluate different body image components...” Missing the studies reference.

“Measurement of the body image is challenging, as it requires instruments capable of capturing body valuation from the individual's perception and/or attitudes.” Missing the reference.

“However, studies that evaluate body image usually do not include orofacial …” Which studies?

“OES assesses satisfaction with orofacial appearance and PIDAQ assesses the impact that dental esthetics have on the individual's life, such as self-confidence, social impact, psychological impact, and concern about the appearance of teeth.” Missing the reference.

Methods

The information about the measuring scales and sample characterization would be clearer if they were present in tables.

Discussion

An English grammar review is required throughout the manuscript.

Conclusion

At the beginning of the conclusion, the authors describe the results. Authors should better structure the conclusion without repeating the results found by the study.

Reviewer #2: Dear authors, thank you for the submission. The manuscript is well-written and well-structured, however, there are some points that must be improved, described below.

L.69-70: “These behaviors can be enhancers for the onset of mental disorders such as eating disorders [2]” What other disorders are related to the behaviors mentioned? Contextualize with what was reported.

L.112-114: “no studies have been published that evaluate this relationship considering both body and facial appearance simultaneously” This is a poor justification of your study. I recommend improving the justification with precise arguments. It seems to me that the narrative described in the previous paragraph is configured as an adequate justification.

The methods performed are adequate and realized according to validated questionnaires. However, it was mentioned that one item of ABS assess was excluded and not considered for mean score calculation. Could this exclusion be considered some bias for calculating the mean, and consequently affect the result?

L.509-10: “Individuals with higher BMI values had higher levels of the negative components of body image and lower levels of the positive components” Please, insert the classification of BMI in the Methods section.

L.526-28: “Thus, we expect that the results of this study could help health care professionals to develop and improve a holistic view of human subjects, which would assist the development of patient-centered treatment plans” This is an important aspect that is absent in the discussion topic. Consider inserting the impact of your findings on the population’s quality of life and the inclusion of public policies or the role of health care for these individuals.

6. PLOS authors have the option to publish the peer review history of their article (what does this mean?). If published, this will include your full peer review and any attached files.

Reviewer #1: No

Reviewer #2: No

---

## [Author Response · Author response to Decision Letter 0]

7 Aug 2022

Dear Dr. Manoelito Ferreira Silva Junior,

Academic Editor PLOS ONE

Thank you for reviewing our manuscript and considering the study for publication after the requested review. We are submitting a revised manuscript highlighting the changes and a clean version. Please kindly see below our response point-by-point to reviewers’ comments and suggestions.

We hope that the responses and the revised manuscript address the reviewers' concerns.

Sincerely,

The Authors

5. Review Comments to the Author

Reviewer #1

•Abstract 

Authors should it clear which form of evaluation, if it occurred through questionnaires. 

-Response: We added this information in the Abstract (methods).

•In the results, the authors should add the real value of p in the sentence: “For men, the affective and satisfaction components of BI had a greater impact on life satisfaction than the other factors. For women, both BI and OA had a similar impact.” 

-Response: We added to the abstract information regarding the impact value (β) and p-value.

•Introduction

The introduction is too long.

“…studies aiming to evaluate different body image components...” Missing the studies reference.

“Measurement of the body image is challenging, as it requires instruments capable of capturing body valuation from the individual's perception and/or attitudes.” Missing the reference.

“However, studies that evaluate body image usually do not include orofacial …” Which studies?

“OES assesses satisfaction with orofacial appearance and PIDAQ assesses the impact that dental esthetics have on the individual's life, such as self-confidence, social impact, psychological impact, and concern about the appearance of teeth.” Missing the reference. 

-Response: We understand that the introduction may be a bit long, however, our manuscript involves concepts that cross areas of knowledge related to body image, dentistry, psychology, and psychometrics. Since Plos One has a broad readership from many different fields of knowledge, we believe it is important to point out some main concepts that may not be usual for all readers. 

We added the references in the sentences that the reviewer pointed out.

•Methods

The information about the measuring scales and sample characterization would be clearer if they were present in tables. 

-Response: Thank you for the suggestion. Regarding the sample characterization, Table 1 presented in the Results already contained most of the variables. To avoid overlapping information by adding a new table with this same information, we just included all the characterization variables (and their results) in Table 1 and cited this table in Methods - Sample Characterization.

Regarding the measured scales, we have removed the psychometric indicators of each scale for each sample from the text (Methods - Scale Measurements). These data have now been presented as supporting information (S1 Table), which also includes the dimension measured by each instrument.

•Discussion

An English grammar review is required throughout the manuscript. 

-Response: The language of the manuscript has been revised by Semantix (project number 76018646).

•Conclusion

At the beginning of the conclusion, the authors describe the results. Authors should better structure the conclusion without repeating the results found by the study. 

-Response: The conclusion was restructured.

Reviewer #2

•Dear authors, thank you for the submission. The manuscript is well-written and well-structured, however, there are some points that must be improved, described below. 

-Response: Thank you. We changed the issues pointed out by the reviewers to improve the manuscript and make it suitable for publication. We hope we have met them.

•L.69-70: “These behaviors can be enhancers for the onset of mental disorders such as eating disorders [2]” What other disorders are related to the behaviors mentioned? Contextualize with what was reported. 

-Response: The sentence was rewritten adding other mental disorders and contextualizing the relationship.

•L.112-114: “no studies have been published that evaluate this relationship considering both body and facial appearance simultaneously” This is a poor justification of your study. I recommend improving the justification with precise arguments. It seems to me that the narrative described in the previous paragraph is configured as an adequate justification. 

-Response: We rewrite the sentence to describe a justification using the argument presented in the previous paragraph.

•The methods performed are adequate and realized according to validated questionnaires. However, it was mentioned that one item of ABS assess was excluded and not considered for mean score calculation. Could this exclusion be considered some bias for calculating the mean, and consequently affect the result? 

-Response: Thank you. We understand the reviewer's concern. But it is actually the maintenance of this item that could be considered a bias in obtaining a score for attention to body shape (measured by ABS). This is because, after conducting Confirmatory Factor Analysis on the study sample data (results in S1 Table), this item presented low factor loading. In other words, this item presents a possible local fit problem, suggesting that the measured construct (attention to body shape) is not reflected in this item for the study sample. We have therefore decided to exclude this item in order to ensure that the estimates obtained are valid and reliable.

In the manuscript (Methods - Measurement Scales - 1st paragraph), we added a sentence justifying that the exclusion of the item was to avoid bias and ensure data validity and reliability.

•L.509-10: “Individuals with higher BMI values had higher levels of the negative components of body image and lower levels of the positive components” Please, insert the classification of BMI in the Methods section. 

-Response: Based on previous studies that evaluated the impact of BMI on well-being (Swami et al. Positive body image is positively associated with hedonic (emotional) and eudaimonic (psychological and social) well-being in British adults. J Soc Psychol. 2018;158(5):541-52. Frederick et al. Correlates of appearance and weight satisfaction in a U.S. National Sample: Personality, attachment style, television viewing, self-esteem, and life satisfaction. Body Image. 2016;17:191-203. Davis et al. The role of body image in the prediction of life satisfaction and flourishing in men and women. J Happiness Stud. 2020;21(2):505-24.), we clarify that in the present study we chose to use BMI quantitatively (Kg/m2) and not considering categories obtained by classification. 

We added a sentence at the end of the last paragraph of the Methods-Sample Characterization to clarify this information.

•L.526-28: “Thus, we expect that the results of this study could help health care professionals to develop and improve a holistic view of human subjects, which would assist the development of patient-centered treatment plans” This is an important aspect that is absent in the discussion topic. Consider inserting the impact of your findings on the population’s quality of life and the inclusion of public policies or the role of health care for these individuals. 

-Response: Thank you for the suggestion. We added a paragraph discussing what the reviewer pointed out.

---

## [Decision Letter · Decision Letter 1]

22 Sep 2022

Impact of body and orofacial appearance on life satisfaction among Brazilian adults

PONE-D-22-14345R1

Dear Dr. Joao Marôco,

We’re pleased to inform you that your manuscript has been judged scientifically suitable for publication and will be formally accepted for publication once it meets all outstanding technical requirements.

Kind regards,

Manoelito Ferreira Silva Junior, Ph.D.

Academic Editor

PLOS ONE

Additional Editor Comments (optional):

I thank the authors for the effort to meet all the points of the reviewers, which substantially improved the quality of the manuscript.

Therefore, we consider the article suitable for publication in its current format.

Best Regards.

Reviewers' comments:

Reviewer's Responses to Questions

**Comments to the Author**

1. If the authors have adequately addressed your comments raised in a previous round of review and you feel that this manuscript is now acceptable for publication, you may indicate that here to bypass the “Comments to the Author” section, enter your conflict of interest statement in the “Confidential to Editor” section, and submit your "Accept" recommendation.

Reviewer #1: All comments have been addressed

Reviewer #2: All comments have been addressed

2. Is the manuscript technically sound, and do the data support the conclusions?

Reviewer #1: Yes

Reviewer #2: Yes

3. Has the statistical analysis been performed appropriately and rigorously? 

Reviewer #1: Yes

Reviewer #2: Yes

4. Have the authors made all data underlying the findings in their manuscript fully available?

Reviewer #1: Yes

Reviewer #2: Yes

5. Is the manuscript presented in an intelligible fashion and written in standard English?

Reviewer #1: Yes

Reviewer #2: Yes

6. Review Comments to the Author

Reviewer #1: (No Response)

Reviewer #2: Dear authors, congratulations for your study. The reviews are adequate, so I'm considering this study for publication.

7. PLOS authors have the option to publish the peer review history of their article (what does this mean?). If published, this will include your full peer review and any attached files.

Reviewer #1: No

Reviewer #2: No

---

## [Editor Report · Acceptance letter]

27 Oct 2022

PONE-D-22-14345R1 

Impact of body and orofacial appearance on life satisfaction among Brazilian adults 

Dear Dr. Marôco:

I'm pleased to inform you that your manuscript has been deemed suitable for publication in PLOS ONE. Congratulations! Your manuscript is now with our production department. 

Kind regards, 

on behalf of

Dr. Manoelito Ferreira Silva Junior 

Academic Editor

PLOS ONE